# Improving Sequence Generative Adversarial Networks with Feature Statistics Alignment

## Abstract

Generative Adversarial Networks (GAN) are facing great challenges in synthesizing sequences of discrete elements, such as mode dropping and unstable training. The binary classifier in the discriminator may limit the capacity of learning signals and thus hinder the advance of adversarial training. To address such issues, apart from the binary classification feedback, we harness a Feature Statistics Alignment (FSA) paradigm to deliver fine-grained signals in the latent high-dimensional representation space. Specifically, FSA forces the mean statistics of the fake data distribution to approach that of real data as close as possible in a finite-dimensional feature space. Experiments on synthetic and real benchmark datasets show the superior performance in quantitative evaluation and demonstrate the effectiveness of our approach to discrete sequence generation. To the best of our knowledge, the proposed architecture is the first that employs feature alignment regularization in the Gumbel-Softmax based GAN framework for sequence generation.

## 1 Introduction

Unsupervised sequence generation is the cornerstone for a plethora of applications, such as machine translation (Wu et al., 2016), image captioning (Anderson et al., 2018), and dialogue generation (Li et al., 2017). The most common approach to autoregressive sequence modeling is maximizing the likelihood of each token in the sequence given the previous partial observation. However, using maximum likelihood estimation (MLE) for sequence modeling is inherently prone to the exposure bias problem (Bengio et al., 2015), which results from the discrepancy between the training and inference stage: the generator predicts the next token conditioned on its previously generated ones during inference but conditioned on its prefix ground-truth tokens during training, yielding accumulative mismatch along with the increment of generated sequence length.

Generative Adversarial Networks (GANs) (Goodfellow et al., 2014) can serve as an alternative to models trained by MLE, which have achieved promising results in generating sequences of discrete elements, in particular, language sequences (Kusner & Hernández-Lobato, 2016; Yu et al., 2017; Lin et al., 2017; Guo et al., 2018; Fedus et al., 2018; Nie et al., 2019; de Masson d'Autume et al., 2019; Zhou et al., 2020; Scialom et al., 2020). GANs consist of two competing networks: a discriminator that is trained to distinguish the generated samples from real data, and a generator that aims to generate high-quality samples to fool the discriminator.

Although having succeeded in avoiding exposure bias issues, GANs still suffer from some intrinsic problems, such as mode dropping, reward sparsity, and training instability. To enrich the informativeness of the discriminator's training signal, several approaches have been proposed by measuring the latent features, such as feature distribution matching (Zhang et al., 2017; Chen et al., 2018) and comparative discriminators (Lin et al., 2017; Zhou et al., 2020). Zhang et al. (2017) and Chen et al. (2018) leveraged feature matching mechanism by minimizing the kernel-based moment-matching metric, such as Maximum Mean Discrepancy and Earth-Mover's Distance, between encoded features. However, merely adopting feature matching in lieu of the original learning signal may lack some guiding feedback at the initial stage of training.

Another approach is to compare the finite latent features with comparative discriminators like ranker and relativistic discriminator. Lin et al. (2017) maintained that the binary classification in the discriminator network limits the learning capacity of tasks because the diversity and richness are circumscribed by the degenerated distribution. RankGAN (Lin et al., 2017) replaced the binary clas-

Figure 1: (a) Standard GANs using a binary classifier as its discriminator; (b) GANs with Feature Statistics Alignment and relativistic discriminator that provide more instructive signals for updating the generator.

sifier with a pairwise feature ranker by comparing the similarities between sample features in the latent space. SAL (Zhou et al., 2020) classified the encoded features of constructed pairwise training examples into three categories, *i.e.*, better / worse / indistinguishable. Nevertheless, adopting a comparative discriminator could provide the fine-grained signals for updating the generator network and may require some coarse credits for further improvements.

In this work, we propose to improve the GANs for sequence generation by jointly considering both the feature statistics matching and relativistic discriminator to serve as fine-grained and coarse learning signals respectively. We leverage the Feature Statistics Alignment (FSA) paradigm to embed the latent feature representations in a finite feature space and force the distribution of generated samples to approach the real data distribution by minimizing the distance between their respective feature representation centroids. Intuitively, matching the mean feature representations of fake and real samples could make the two data distributions closer. Besides, the relativistic discriminator (Jolicoeur-Martineau, 2019) is employed to measure the comparative information between generated and real sequences and empirically to show the effectiveness during the model training.

Our experimental results illustrate the effectiveness of FSA techniques and large batch size to alleviate the gradient vanishing problem and stabilize the training process in comparison with the vanilla Gumbel-Softmax GANs. Besides, our models could generate discrete text sequences with high quality in terms of the semantic coherence and grammatical correctness of language, as evaluated with crowdsourcing. Furthermore, we empirically demonstrate that the proposed architecture overshadows most existing models in terms of quantitative and qualitative evaluation. To the best of our knowledge, the proposed framework is the first to adopt the statistics feature alignment paradigm in the Gumbel-Softmax based GAN framework for discrete sequence generation.

## 2    ADVERSARIAL SEQUENCE GENERATION

Adversarial sequence generation has attracted broad attention for its properties to solve the exposure bias issue suffered with maximum likelihood estimation (MLE) for generating language sequences. Based on the game theory, its goal is to train a generator network $G(\boldsymbol{z}; \boldsymbol{\theta}^{(G)})$ that produces samples from the data distribution $p_{\text{data}}(\boldsymbol{x})$ by decoding the randomly initialized noise $\boldsymbol{z}$ (*i.e.*, standard normal distribution) into the sequence $\boldsymbol{x} = G(\boldsymbol{z}; \boldsymbol{\theta}^{(G)})$, where the training signal is provided by the discriminator network $D(\boldsymbol{x}; \boldsymbol{\phi}^{(D)})$ that is trained to distinguish between the samples drawn from the real data distribution $p_{\text{data}}$ and those produced by the generator. The minimax objective of adversarial training is formulated as:

$$\min_{\boldsymbol{\theta}^{(G)}} \max_{\boldsymbol{\phi}^{(D)}} \mathbb{E}_{\boldsymbol{x} \sim p_{\text{data}}} \big[ \log D(\boldsymbol{x}; \boldsymbol{\phi}^{(D)}) \big] + \mathbb{E}_{\boldsymbol{z} \sim p_{\boldsymbol{z}}} \big[ \log \big( 1 - D_{\boldsymbol{\phi}^{(D)}}(G(\boldsymbol{z}; \boldsymbol{\theta}^{(G)})) \big) \big]. \tag{1}$$

Despite the impressive results of GANs in the sequence generation (Yu et al., 2017; Gulrajani et al., 2017; Scialom et al., 2020), there are still several fundamental issues in the GAN training: (a) Training instability, which arises from the intrinsic nature of minimax games in GANs; (b) Mode dropping, which is the fact that GANs only generate samples with limited patterns in the real data distribution instead of attending to diverse patterns (Chen et al., 2018); (c) Reward sparsity, which is because that it is easier to train the discriminator than the generator, making it difficult to acquire the instructive feedback (Zhou et al., 2020).

Due to the non-differentiability of gradients caused by sampling operations between the generator and discriminator for sequence generation, the majority of previous works have resorted to reinforcement learning (RL) heuristics with Monte Carlo search to collect the credits from the discriminator. The usage of RL may further deteriorate the instability of model training and exacerbate the reward sparsity problem. Gumbel-Softmax relaxation has proven to be an alternative to RL techniques (Kusner & Hernández-Lobato, 2016; Nie et al., 2019). How to efficiently train GANs with the Gumbel-Softmax trick still remains under-explored. Therefore, we utilize the Gumbel-Softmax reparameterization instead of conventional policy gradients in our framework.

## 3 METHODOLOGY

As illustrated in Fig. 1, standard GANs employ the real/fake binary classifier as the discriminator, which is prone to be overtrained in comparison with the generator (Salimans et al., 2016). To prevent the discriminator from overfitting and further stabilize the training process, we propose to leverage the FSA techniques and relativistic discriminator by comparing the fake and real distributions from two different aspects.

Compared with conventional GANs, the proposed framework enjoys the following advantages: (a) In the earlier training stage, the relativistic discriminator could estimate the probability that how much better the real data is in comparison with the generated samples. This could provide the relatively "coarse" credits to the generator. (b) When it comes to the later training stage and the quality of generated samples becomes high, FSA measures the "fine-grained" difference between latent features of fake and real data batches, precluding the discriminator from being overly confident. In other words, the FSA can be regarded as a kind of regularization for adversarial training.

### 3.1 FEATURE STATISTICS ALIGNMENT

Intuitively, aligning the statistics of embedded feature representations increases the opportunities to capture the various modes of the data distribution. For brevity, we utilize the first-order mean statistics in our framework and leave the higher-order statistics for future work.

Denoting the feature extractor as $F_\omega$ parameterized by $\omega$, a minibatch of real data samples as $\boldsymbol{x}$ with the batch size of $N$, we propose two variants of FSA formulations, which calculate the mean squared difference and Euclidean distance between the minibatch centroids of real and fake feature representations. To reduce the parameter amount, we share the weights between the discriminator and feature extractor of FSA.

**Mean Squared Alignment (MSA)**   We take the mean squared difference between the centroids of fake and generated distributions as the feature alignment metric:

$$\mathcal{L}_{\text{MSA}} = \left\| \mathbb{E}_{\boldsymbol{x} \sim p_{\text{data}}} \left[ F_\omega(\boldsymbol{x}) \right] - \mathbb{E}_{\boldsymbol{z} \sim p_{\boldsymbol{z}}} \left[ F_\omega(G(\boldsymbol{z}; \boldsymbol{\theta}^{(G)})) \right] \right\|_2^2 \tag{2}$$

$$= \left\| \frac{1}{N} \sum_{i=1}^{N} F_\omega(\boldsymbol{x}_i) - \frac{1}{N} \sum_{i=1}^{N} F_\omega(G(\boldsymbol{z}_i; \boldsymbol{\theta}^{(G)})) \right\|_2^2. \tag{3}$$

**Mean Distance Alignment (MDA)**   Another intuitive approach is to calculate the distance between two sample centroids, which is equivalent to the square root of MSA mathematically:

$$\mathcal{L}_{\text{MDA}} = \left\| \mathbb{E}_{\boldsymbol{x} \sim p_{\text{data}}} \left[ F_\omega(\boldsymbol{x}) \right] - \mathbb{E}_{\boldsymbol{z} \sim p_{\boldsymbol{z}}} \left[ F_\omega(G(\boldsymbol{z}; \boldsymbol{\theta}^{(G)})) \right] \right\|_2 \tag{4}$$

$$= \sqrt{L_{\text{MSA}}}. \tag{5}$$

By forcing the mean statistics of fake data to be close to the real samples, the generator could receive more informative signals in the training process. It is worth noting that the large batch size is helpful to reduce the variance of small mini-batches.

## 3.2 RELATIVISTIC DISCRIMINATOR

We consider the Relativistic Discriminator (Jolicoeur-Martineau, 2019) to take into account the relative confidence that the given real data is more realistic than the randomly sampled fake data. In the standard GAN, defining the discriminator as $D(\boldsymbol{x}) = \text{sigmoid}(H(\boldsymbol{x}))$, where $H(\cdot)$ represents the non-transformed layer before the final non-linearity. The objectives for the discriminator and generator in terms of the Relativistic Discriminator are defined as:

$$\mathcal{L}_{\text{RD}} = -\mathbb{E}_{\boldsymbol{x} \sim p_{\text{data}}, \boldsymbol{z} \sim p_{\boldsymbol{z}}}[\log a(H(\boldsymbol{x}) - H(G(\boldsymbol{z}; \boldsymbol{\theta}^{(G)})))], \tag{6}$$

$$\mathcal{L}_{\text{RG}} = -\mathbb{E}_{\boldsymbol{x} \sim p_{\text{data}}, \boldsymbol{z} \sim p_{\boldsymbol{z}}}[\log a(H(G(\boldsymbol{z}; \boldsymbol{\theta}^{(G)})) - H(\boldsymbol{x}))], \tag{7}$$

where $a$ represents the activation function to be relativistic (we use sigmoid function in our experiments), RD and RG denote the loss terms for the discriminator and generator respectively.

## 3.3 OVERALL TRAINING OBJECTIVES

Previous works using moment matching techniques to support the training on both the discriminator and generator (Zhang et al., 2017; Chen et al., 2018). However, the generator is always more difficult to train than the discriminator, resulting in the training instability and reward sparsity. To relieve these issues, we only adopt the FSA techniques to enhance the generator but keep the objective of the discriminator unchanged. This could pass more informative signals only to the generator, and also prevent the discriminator to be overtrained.

Therefore, the overall training objectives for the proposed framework are defined as:

$$\mathcal{L}_{\text{D}} = \mathcal{L}_{\text{RD}}, \tag{8}$$

$$\mathcal{L}_{\text{G}} = \mathcal{L}_{\text{RG}} + \mathcal{L}_{\text{FSA}}, \tag{9}$$

where $\mathcal{L}_{\text{FSA}}$ takes the form of $\mathcal{L}_{\text{MSA}}$ and $\mathcal{L}_{\text{MDA}}$ as Eq. 2 and Eq. 4.

The goal of the discriminator is to maximize the gap between the generated and real data, whereas the generator jointly considers two different aspects simultaneously: it not only competes with the discriminator by maximizing the gap in terms of the relativistic signals but takes into account the additional leaked feature information from the discriminator. The idea of leaked features from the discriminator is similar to LeakGAN (Guo et al., 2018). The FSA term on the RHS can also be regarded as a dynamic regularizer for the sequence generator.

## 3.4 TRAINING WITH DISCRETE SEQUENCE

### 3.4.1 GUMBEL-SOFTMAX DISTRIBUTION

Conventional GANs for generating discrete sequences are inherently unable to backpropagate the gradient through samples due to the non-differentiable sampling from a categorical distribution. Gumbel-Softmax distribution (Jang et al., 2017; Maddison et al., 2017) was proposed to deal with this issue by smoothly annealed to approximate the categorical distribution.

Denoting the output probabilities $\pi_1, \pi_2, \cdots, \pi_{|V|}$, where $|V|$ represents the output vocabulary size in the generator, the Gumbel-Max trick (Maddison et al., 2014) can be parameterized as:

$$y_i = \text{one\_hot}(\arg\max_i[g_i + \log \pi_i]) \quad \text{for } i = 1, \cdots, |V|, \tag{10}$$

where $\{g_i | i = 1, \cdots, |V|\}$ are i.i.d from the Gumbel(0,1) distribution, that is, $g_i = -\log(-\log u_i)$ with $u_i$ is drawn from a standard uniform distribution Uniform(0,1). one\_hot represents the $|V|$-dimensional one hot encoding.

Gumbel-Softmax approximates the non-differential $\arg\max$ operation using the softmax function:

$$\hat{y}_i = \frac{\exp((\log(\pi_i) + g_i)/\tau)}{\sum_{j=1}^{|V|} \exp((\log(\pi_j) + g_j)/\tau)}, \quad \text{for } i = 1, \cdots, |V|, \tag{11}$$

where $\tau$ denotes the softmax temperature to modulate the exploitation and exploration during training. When $\tau$ approaches too high, the approximation is nearly equiprobable, encouraging the generator to explore different options. In contrast, the lower $\tau$ could discourage the exploration and tend to exploit during training. In particular, when $\tau \to 0$, $\hat{y}_i$ approaches the result of one hot operator as in Eq. 10, whereas $\hat{y}_i$ will degenerated into a uniform distribution when $\tau \to \infty$.

### 3.4.2 Architecture and Adversarial Training

**RNN Generator** Due to the free-running mode of GAN's generator, it is unsuitable to adopt the transformer-based models due to the non-recurrence nature. Thus, we investigate the Recurrent Neural Network (RNN) based models, such as Long Short-Term Memory (LSTM) (Hochreiter & Schmidhuber, 1997), and Relational Memory Core (RMC) (Santoro et al., 2018).

**CNN Discriminator & Feature Extractor** We use the convolutional neural networks (CNN) architecture (Kim, 2014) as the discriminator and feature extractor for input sequences. We employ the multi-channel convolution using multiple filters with various window sizes to extract the distinct n-gram features, followed by a max-over-time pooling operation to gather the most salient features, *i.e.*, features with the highest value for each feature map.

**Adversarial Training Algorithm** Alg. 1 illustrates the overall training process of the proposed framework. The Relativistic Discriminator and the generator could reach the Nash Equilibrium when the generator could fool the discriminator into accepting its output as being true. Since the discriminator is easy to be overtrained, we do not pretrain the discriminator but only pretrain the generator using MLE for few epochs.

---

**Algorithm 1** Adversarial Training with Feature Statisitcs Alignment

---

1: **Require:** generator $G_\theta$; discriminator $D_\phi$; samples of real data $\mathbb{S}$; generator training step $g$; discriminator training step $k$; the generator pretraining epochs $m$.
2: Pretrain $G_\theta$ using MLE on $\mathbb{S}$ for $m$ epochs
3: **repeat**
4:    **for** $g$ steps **do**
5:       Sample a minibatch from real data $\mathbb{S}$
6:       Generate a minibatch of samples $\boldsymbol{x}_g \sim G_\theta$
7:       Update $G_\theta$ via Eq.(9)
8:    **end for**
9:    **for** $k$ steps **do**
10:       Sample a minibatch from real data $\mathbb{S}$
11:       Sample a minibatch from the generated data
12:       Train the discriminator $D_\phi$ by Eq.(8)
13:    **end for**
14: **until** convergence

---

## 4 Experiments

### 4.1 Experimental Setting

Similar to (Lin et al., 2017; Guo et al., 2018; Nie et al., 2019; Zhou et al., 2020), we evaluate the proposed framework based on the Texygen benchmark platform (Zhu et al., 2018) for adversarial text generation. Experiments were conducted on the synthetic and real datasets: (a) synthetic data, which is generated by an oracle single-layer LSTM as in (Yu et al., 2017); (b) MS COCO Image Caption dataset (Chen et al., 2015); (c) EMNLP WMT 2017 News dataset (Guo et al., 2018). Table 1 summarizes the statistics of benchmark datasets for evaluation.

Table 1: Summary of experimental datasets.

| dataset | vocabulary size | sequence length | training set | test set |
|---|---|---|---|---|
| synthetic data | 5,000 | 20 / 40 | 10,000 | 10,000 |
| MS COCO | 4,657 | 37 | 10,000 | 10,000 |
| EMNLP2017 WMT News | 5,255 | 51 | 27,8586 | 10,000 |

For the synthetic data experiments, we utilize a single-layer LSTM initialized by the standard normal distribution as the oracle model, which is used to generate 10,000 samples of length 20 and 40

respectively as the real samples. We use the negative log-likelihood (NLL) under the oracle data distribution for evaluation, termed $NLL_{oracle}$. For the real data experiments, the BLEU score (Papineni et al., 2002) serves as a metric to evaluate the n-gram statistics overlapping on the whole dataset.

To measure the diversity of generated samples, the NLL of the generator (denoted as $NLL_{gen}$) is used by computing the NLL of reference samples in the test set by the generator. Considering that BLEU scores always focus on the local text statistics and may be insufficient for evaluating the overall quality of texts, we conducted additional human evaluation via crowdsourcing on the generated samples of all comparison models. See Appendix A for more experimental details.

We compare the proposed framework with the MLE baseline and other state-of-the-art models, involving SeqGAN (Yu et al., 2017), RankGAN (Lin et al., 2017), LeakGAN (Guo et al., 2018), RelGAN (Nie et al., 2019), and Self-Adversarial Learning (SAL) (Zhou et al., 2020).

### 4.2 EXPERIMENTAL RESULTS

#### 4.2.1 SYNTHETIC DATA

Table 2 illustrates the performance of different models on $NLL_{oracle}$. Since our experiments achieved better results using MSA instead of MDA on synthetic data, we only report the optimal results with MSA in Table 2. Our models outperform other prevalent adversarial models in terms of the generated sample quality, demonstrating the effectiveness of our proposed method.

In practice, we found that the LSTM generator could outperform the RMC generator for producing the sequence with the length of 20, whereas RMC generators exceed LSTMs for long sequence generation. This may be due to the fact that LSTMs may forget the long-term dependencies with the sequence length increases, but the self-attention based relational memory cell in RMC could mitigate the issues using interactive memory slots. This demonstrates that the proposed model could produce samples with high quality. As to the diversity, our method achieves competitive $NLL_{gen}$ score compared with baselines models (see Appendix C.1 for the detail).

Table 2: The $NLL_{oracle}$ performance of different models ($\tau = 1$) on the synthetic dataset with the sequence length of 20 and 40 respectively. For NLL, the lower, the better.

| Length | MLE | SeqGAN | RankGAN | LeakGAN | RelGAN | SAL | Ours (LSTM) | Ours (RMC) | Real |
|---|---|---|---|---|---|---|---|---|---|
| 20 | 9.038 | 8.736 | 8.247 | 7.038 | 6.680 | 7.71 | **5.047** | 5.819 | 5.750 |
| 40 | 10.411 | 10.310 | 9.958 | 7.191 | 6.765 | 9.31 | 5.909 | **5.087** | 4.071 |

#### 4.2.2 MS COCO DATASET

To further test the performance on the real data, we run and evaluate our model on MS COCO image caption datasets. The data and preprocessing remain the same as in Texygen (Zhu et al., 2018). Empirically, we found that RMC achieves better results in terms of the long sequences, and thus reports the results using the RMC generator if not otherwise specified.

Table 3: BLEU and $NLL_{gen}$ performance on MS COCO image captions with $\tau = 0.1$ for the proposed models with MSA and MDA. For BLEU scores, the higher, the better.

| Model | BLEU-2 | BLEU-3 | BLEU-4 | BLEU-5 | $NLL_{gen}$ |
|---|---|---|---|---|---|
| MLE | 0.731 | 0.497 | 0.305 | 0.189 | 0.718 |
| SeqGAN | 0.745 | 0.498 | 0.294 | 0.180 | 1.082 |
| RankGAN | 0.743 | 0.467 | 0.264 | 0.156 | 1.344 |
| LeakGAN | 0.746 | 0.528 | 0.355 | 0.230 | **0.679** |
| RelGAN | 0.849 | 0.687 | 0.502 | 0.331 | 0.756 |
| SAL | 0.785 | 0.581 | 0.362 | 0.227 | 0.873 |
| Ours (MSA) | **0.959** | **0.866** | **0.759** | **0.630** | 0.760 |
| Ours (MDA) | 0.938 | 0.863 | 0.731 | 0.582 | 0.717 |

Table 3 exhibits the final results of the BLEU and $NLL_{gen}$ scores on different comparison models. Our models reported in Table 3 set the softmax temperature as 0.1 for the model with MSA and that with MDA. Notably, our model shows the significant improvement on previous methods, consistently overshadowing the state-of-the-art models in terms of the sample quality (indicated by BLEU scores) while maintaining the diversity (indicated by $NLL_{gen}$).

### 4.2.3 EMNLP2017 WMT NEWS DATASET

Table 4 presents the considerable improvements of our model (w.r.t. quality) on EMNLP2017 WMT News dataset, with the temperature of 1 for models with MSA and MDA. The maximum length in the EMNLP2017 dataset is 51, greatly challenging the generation task.

It can be observed that only RankGAN and LeakGAN slightly outrank the MLE in terms of the $NLL_{gen}$, which may be due to the fact that feature ranking or leakage information could pass more internal information from the discriminator to the generator, and thereby smoothly assist the training process of the generator. By leveraging the FSA methods, our model greatly outperforms these models and yielding the long sequences with promising qualities.

Table 4: The BLEU and $NLL_{gen}$ performance on EMNLP2017 WMT News dataset with temperatures of 1 for the proposed models with MSA and MDA.

| Model | BLEU-2 | BLEU-3 | BLEU-4 | BLEU-5 | $NLL_{gen}$ |
|---|---|---|---|---|---|
| MLE | 0.768 | 0.473 | 0.240 | 0.126 | **2.382** |
| SeqGAN | 0.777 | 0.491 | 0.261 | 0.138 | 2.773 |
| RankGAN | 0.727 | 0.435 | 0.209 | 0.101 | 3.345 |
| LeakGAN | 0.826 | 0.645 | 0.437 | 0.272 | 2.356 |
| RelGAN | 0.881 | 0.705 | 0.501 | 0.319 | 2.482 |
| SAL | 0.788 | 0.523 | 0.281 | 0.149 | 2.578 |
| Ours (MSA) | **0.932** | **0.798** | 0.585 | **0.404** | 3.999 |
| Ours (MDA) | 0.916 | 0.784 | **0.592** | 0.386 | 2.732 |

Apart from the automatic quantitative evaluation, we also conducted human evaluation on the MS COCO Image Captioning dataset. We randomly sampled 100 sentences for each model and the real data, then asked 12 different people to score them on the scale of 1-5 with anonymizing the model's identity. Please see Appendix. C.2 for more details of human evaluation.

Our models received the highest score in comparison with other models, where the model with MSA receives higher credits than that with MDA. By manually going through the generated samples, we found that models with MSA tend to generate long sentences with higher quality but run into the mode dropping issue. In contrast, models with MDA posses a better trade-off between the diversity and quality of samples, yielding promising results in comparison with previous models.

Table 5: Mean and standard deviation results of human evaluation w.r.t different models on MS COCO Image Caption dataset. Note that "Real" indicates the real data samples.

| Model | MLE | SeqGAN | RankGAN | LeakGAN | MaliGAN |
|---|---|---|---|---|---|
| Human score | $3.127 \pm 0.124$ | $3.062 \pm 0.100$ | $3.048 \pm 0.127$ | $3.018 \pm 0.104$ | $3.031 \pm 0.123$ |

| Model | TextGAN | RelGAN | Ours (MSA) | Ours (MDA) | Real |
|---|---|---|---|---|---|
| Human score | $1.973 \pm 0.168$ | $3.687 \pm 0.181$ | $4.209 \pm 0.245$ | $3.878 \pm 0.276$ | $3.444 \pm 0.122$ |

### 4.3 DISCUSSION

Fig. 2 (Left) illustrates the ablation test of FSA in terms of the BLEU-4 score (See Appendix D.2 for all the results). It can be observed that the increasing trend for models with FSA outreached that w/o feature alignment mechanism and our models achieve superior performance in the later training stage. The utilization of FSA leads to significant performance gains, as it could provide the consecutive "fined-grained" smoother learning signals to update the generator.

Fig. 3 demonstrates the relative importance of each component of our models with an ablation test. The usage of FSA results in the most significant performance gain, followed by Gumbel-Softmax, and large batch size. A large batch size can boost the performance due to the variance reduction, accurate estimates of feature statistics, and stabilizing effect during the adversarial training (see Fig. 2 (Right)).

As to the Gumbel-Softmax trick, we conduct experiments on various temperature values and find that a fined-tuned temperature hyperparameter could account for the second important factor to contribute to the overall performance boost. Please check Appendix D.3 for the detailed results w.r.t the Gumbel-Softmax temperature.

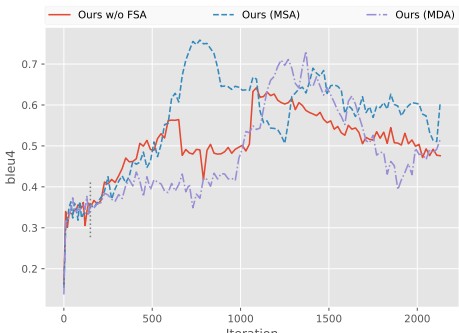 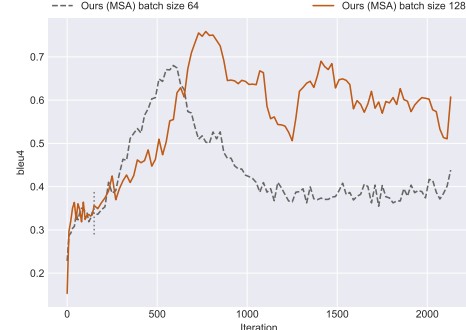

Figure 2: (Left) BLEU-4 scores of the proposed model w/ and w/o FSA mechanism. (Right) BLEU-4 scores of the proposed models with various batch size (64 v.s. 128). The vertical dash lines indicate the end of generator pretraining. MS COCO results unless otherwise specified.

## 5 RELATED WORK

There has been a large category of GANs for sequence generation, which heavily rely on the RL paradigm. SeqGAN (Yu et al., 2017) regards the sequence generation as a Markov decision making process, estimates the rewards via Monte Carlo search, and trains the generator with policy gradient. RankGAN (Lin et al., 2017) and SAL (Zhou et al., 2020) replace the binary classifier in the discriminator as comparative discriminators to take into account the relation between constructed pair samples. MaliGAN (Che et al., 2017) utilizes the information in the discriminator as an additional source of training signals on the MLE objective to reduce the variance of gradients. LeakGAN (Guo et al., 2018) leaks the intermediate feature information via a manager to guide the generator, which is inspired by hierarchical RL. ColdGAN (Scialom et al., 2020) integrates the advance of importance sampling, Proximal Policy Optimization (PPO) algorithm (Schulman et al., 2017), and nucleus sampling (Holtzman et al., 2019) for finetuning the pretrained T5 (Raffel et al., 2019) and BART (Lewis et al., 2019).

Figure 3: Ablation Study of the proposed model.

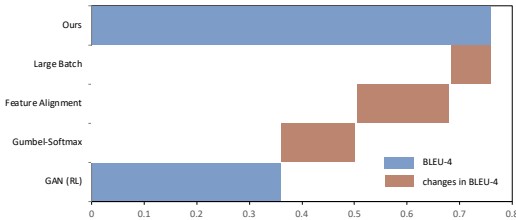

Another approach applies non-RL methods for adversarial sequence generation by either approximating the categorical sampling or directly using the continuous latent representation. TextGAN (Zhang et al., 2017) uses feature matching techniques via a kernelized discrepancy in the Reproducing Kernel Hilbert Space. FMGAN (Chen et al., 2018) proposes to match the feature distributions using a Feature-Mover's Distance. Similar to our proposed model, both of them apply annealed soft-argmax for approximation. ARAML Ke et al. (2019) utilizes Reward Augmented Maximum Likelihood by sampling from the stationary distribution to acquire rewards. However, none of them adopt the Gumbel-Max trick to reparameterize the categorical sampling. Besides, they applied feature matching as the training objectives of both the discriminator and generator, whereas we only apply the feature statistics matching to modulate the generator. Gumbel-Softmax (GS) GAN (Kusner & Hernández-Lobato, 2016) and RelGAN (Nie et al., 2019) prove the effectiveness of Gumbel-Softmax on unsupervised sequence generation. DialogeWAR Gu et al. (2018) employs a GS GAN within the latent variable space for dialogue generation. However, improving the training of GS GANs still remains an open problem. Our model aims to promote the GS GAN with the proposed FSA and other techniques to boost the training of language GANs. We also report a list of other techniques we tried but proved to be unsuccessful or unnecessary in Appendix B.

## 6 CONCLUSION

We propose an adversarial training framework for discrete sequence generation, by leveraging the advance of Feature Statistics Alignment and Gumbel-Softmax relaxation. Our model empirically shows superior performance in terms of the quantitative and human evaluation. In the future, it would be a promising direction to extend the proposed model to conditional text generation, such as text style transfer.

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

# A    EXPERIMENTAL DETAILS

## A.1    TRAINING DETAILS

**Generator**    The input embedding dimension for the generator is set to 32. As for the LSTM generator, we use the single-layer LSTM with the hidden dimension of 32. Then adopt a linear transformation to get the logits at each time step, and iteratively feed the sampled output tokens into the generator at the next time step. As for the RMC generator, we follow the experimental settings as (Nie et al., 2019), setting the memory size as 256, memory slots as 1, attention head number as 2. After the one-layer RMC, a linear projection is applied to get the output logits at each step.

**CNN Discriminator**    The input embedding dimension for the discriminator is set to 64. We adopt the filter size of $\{2, 3, 4, 5\}$ with the number of 300 channels for each. A max-over-time pooling is adopted after the convolution layer. Afterward, a highway layer that is identical to SeqGAN (Yu et al., 2017) is used followed by a linear transformation with the dimension of 100. Finally, apply a linear transformation to get the final logits. The feature extractor for Feature Statistics Alignment shares the identical architecture and weights with the CNN discriminator, and the leaked feature dimension is set to 100.

**Optimization**    We use Adam optimizer with $\beta_1 = 0.9$ and $\beta_2 = 0.999$. The initial learning rate for the generator was set to 1e-2 and 1e-4 for pretraining and adversarial training. We set the initial learning rate as 1e-4 for the discriminator during the adversarial training. To prevent overfitting, we clip the gradients of parameters whose $L_2$ norm exceeds 5.

**Training Procedure**    We conduct experiments to finetune the following experiments: the batch size of $\{32, 64, 128\}$, the Gumbel-Softmax temperature $\tau \in \{1, 0.5, 0.1, 0.01.0.001\}$. The training steps of generator and discriminators are set to $g = 1$ and $d = 5$, respectively. The generator is pretrained for 150 epochs before adversarial training. Finally, the optimal batch size is set to 128 for both synthetic and real datasets. It is worth noting that we also test the batch size to 256, which requires too much GPU resource but do not show obvious improvement.

# B    NEGATIVE RESULTS

Here we list some approaches that we tried but proved unsuccessful:

- Using Mogrifier LSTM as the generator, which achieves similar results as vanilla LSTMs on the synthetic data.
- Using a Wasserstein loss instead of current Relativistic Discriminator. Not as stable as current solutions.
- Using the Transformer model as the discriminator. It achieves unsatisfied results with the current experimental settings.
- Using interleaved training instead of two-stage training, *i.e.,* adversarial training after pretraining. It is unsuccessful to train the generator for 15 iterations after one iteration using MLE.
- Using top-k sampling and nucleus sampling, instead of the argmax in the Gumbel-Max trick. This does not always boost the final performance.
- Using a hinge loss on the discriminator. This did not improve over the current relativistic loss.

# C    EVALUATION DETAILS

## C.1    NLL$_{\text{GEN}}$ ON SYNTHETIC DATA

Table 6 reports the NLL$_{\text{gen}}$ metric of comparison models. It can be seen that the proposed model with RMC generator achieves comparative results in comparison with baselines in terms of both short and long texts.

Table 6: The $\text{NLL}_{\text{gen}}$ performance of different models ($\tau = 1$) on the synthetic dataset with the sequence length of 20 and 40 respectively. For the NLL score, the lower, the better.

| Length | MLE | SeqGAN | RankGAN | SAL | Ours (LSTM) | Ours (RMC) |
|---|---|---|---|---|---|---|
| 20 | 5.96 | 6.61 | 7.14 | 6.58 | 7.73 | 5.12 |
| 40 | 6.55 | 6.98 | 7.05 | 6.97 | 7.59 | 6.89 |

## C.2 HUMAN EVALUATION

Acceptance (*i.e.* if a sentence is acceptable), grammaticality (*i.e.*, if a sentence is grammatically correct), and meaningfulness (*i.e.*, if a sentence makes sense) are three main standards for the text quality evaluation. Please note that any minor text formatting issues which will not negatively influence the understanding and correctness of the sentences (*e.g.*, punctuation, capitalization, spelling errors, extra spaces) can be ignored. Please also note: a sentence consists of less than 10 words should get one point deducted. Table 7 below gives more detailed criteria.

It is worth to mention that the human evaluation is used to measure the quality of generated sentences rather than the diversity.

Table 7: The human evaluation scale from 1 to 5 with corresponding criteria and example sentences.

| Scale | Criterion & Example |
|---|---|
| 5 - Excellent | Grammatical, acceptable, and meaningful. For example, "a man is carving under yellow planes ." |
| 4 – Good | Include 1 to 2 tiny grammatical errors, and the whole sentence is mostly acceptable and meaningful. For example, "two giraffe standing in front of them ." |
| 3 – Fair | Include major grammatical errors, but the whole sentence is still acceptable and making sense. For example, "a kitchen with a grill roll from him ." |
| 2 – Poor | Include severe grammatical errors, and the whole sentence does not make sense, but some parts are still acceptable. For example, "a motorcycle on a paved road on the freeway ." |
| 1 - Unacceptable | It is basically a string of words with random order and totally ungrammatical. The entire sentence does not make any sense. For example, "a city ." |

## C.3 HUMAN EVALUATION ANALYSIS

The model with MSA tends to generate grammatically correct sentences, and the sentences tend to be longer. For example, "A man is sitting on a motorcycle on a busy street, in a city." Though it produces the samples with high quality by human evaluation, however, it does not solve the mode dropping collapse.

In contrast, models with MDA tend to generate a more variety of sentences rather than repeated ones. Most sentences are grammatically correct and meaningful. They follow the SVO sentence structure with Preposition Phrase (PP) placed at the acceptable position in a sentence. Even though some of the auxiliary or main action verbs are missing, the meaning of each sentence can still be understandable and making sense. There is no obvious mode dropping issues according to the generated samples of MDA.

## D DETAILED RESULTS OF ABLATION STUDY

### D.1 IMPACT OF FEATURE STATISTICS ALIGNMENT

See Fig. 4 for the results of the ablation study on FSA techniques.

### D.2 IMPACT OF LARGE BATCH SIZE

Fig. 5 shows the training curve of our model on various batch sizes. It is observed that the increase of batch size could provide the performance boost, due to the variance reduction of gradients and the stability of adversarial dynamics.

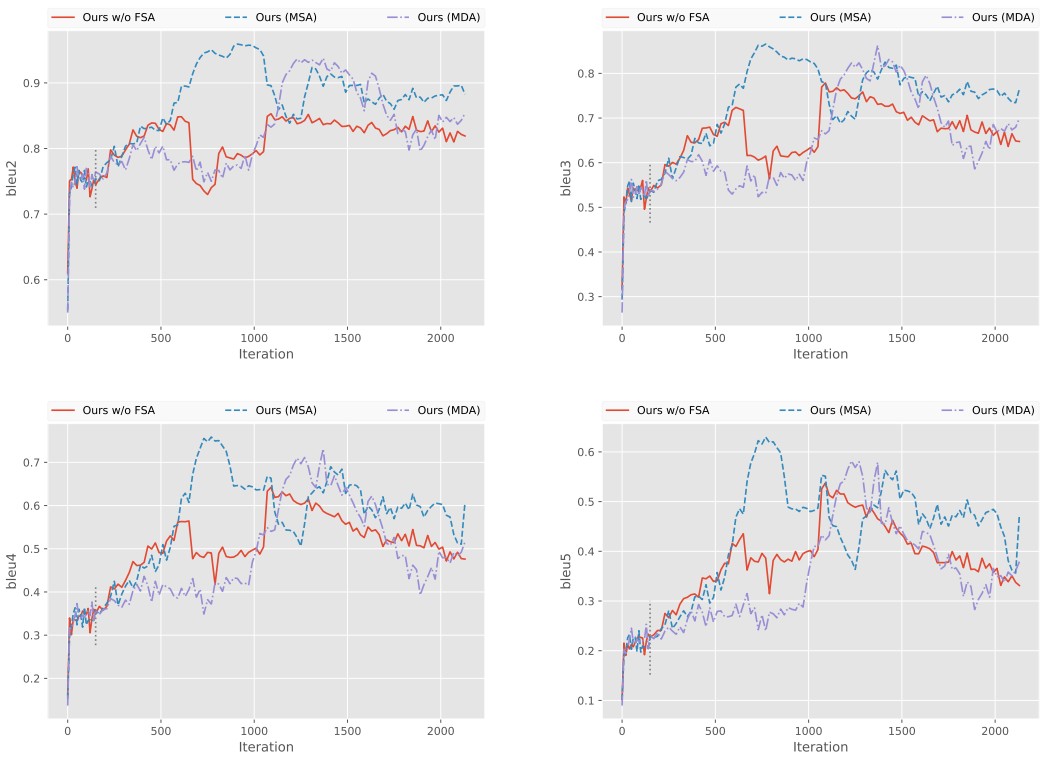

Figure 4: Training curves of BLEU scores on MS COCO Image Caption dataset w/ and w/o FSA mechanism.

### D.3 IMPACT OF GUMBEL-SOFTMAX TEMPERATURE

Fig. 6 reports the BLEU scores with different temperatures on MS COCO dataset. It can be seen that a suitable $\tau$ could greatly advance the automatic evaluation scores.

## E GENERATED SAMPLES ON REAL DATASET

### E.1 GENERATED SAMPLES ON MS COCO DATASET

Table 8 displays samples of generated samples from all baseline models and references on MS COCO dataset.

Table 9 shows the randomly sampled sentences from the proposed models generated on MS COCO Dataset.

### E.2 GENERATED SAMPLES ON EMNLP2017 WMT NEWS DATASET

Table 10 presents the random sampled sentences from our models generated on EMNLP2017 WMT News Dataset.

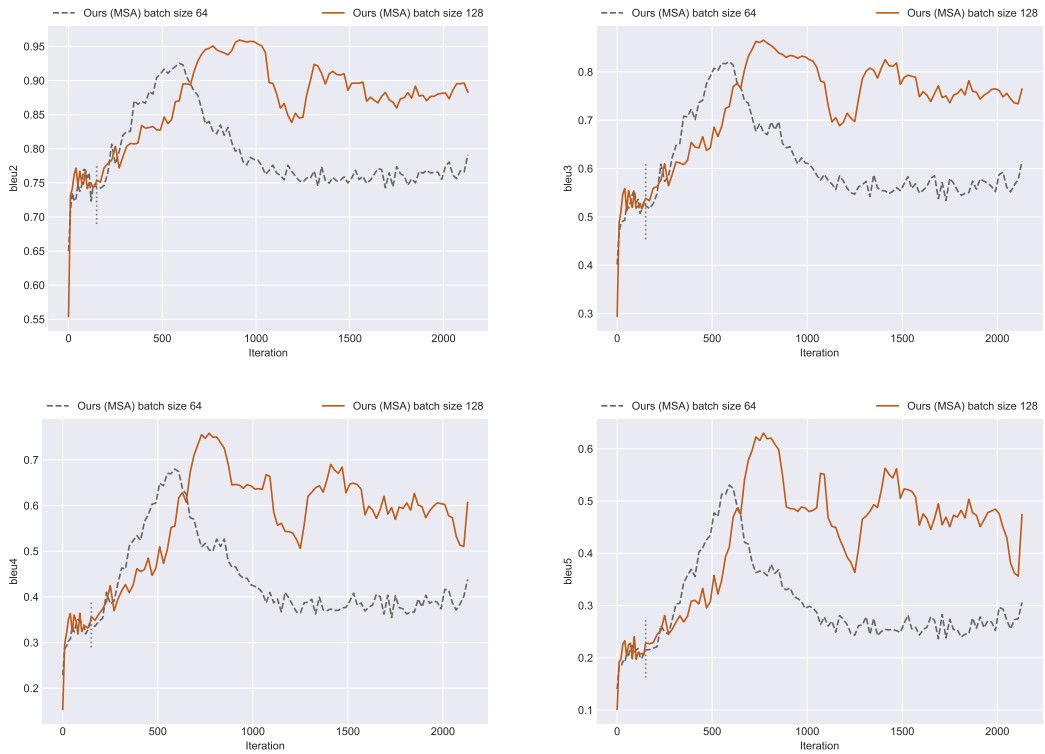

Figure 5: Training curves of BLEU scores on MS COCO Image Caption dataset with various batch sizes.

Table 8: Samples of baseline models and real dataset on MS COCO Image Captioning dataset.

|  | Samples |
|---|---|
| Real | a single kite flies high above a body of water as a person stands on the edge of the water .
a man wearing an apron in an industrial kitchen reaching for a pot . |
| MLE | a man watches on his bike , in a lake on a field .
a women is standing behind an orange table in helmet on a child in the background . |
| SeqGAN | some people sitting on top of luggage near a truck .
a man sitting in a bath tub on tops . |
| TextGAN | a man riding a motorcycle .
a bathroom with a sink , and a table . |
| LeakGAN | a man standing next to her cell phone on a street sign .
a woman is holding a child in the air . |
| MaliGAN | a woman is standing and another oak cake on a drain .
a man standing in a kitchen with her laptop and two tables |
| RankGAN | a colorful bike is is down next to a large mirror .
a man is riding a bike down a track . |
| RelGAN | a woman walking with a dog in the city in front of a city bus .
a man sitting on a bed in a room with a chair on the couch . |
| Ours (MSA) | a man is sitting on a motorcycle on a busy street , in a city .
a man sitting on a motorcycle on a crowded street near a building , with a bicycle in a parking lot . |
| Ours (MDA) | a person is riding a motorcycle on a city street with a woman standing on the back of it .
a man with a woman standing next to a fire hydrant wearing a backpack . |

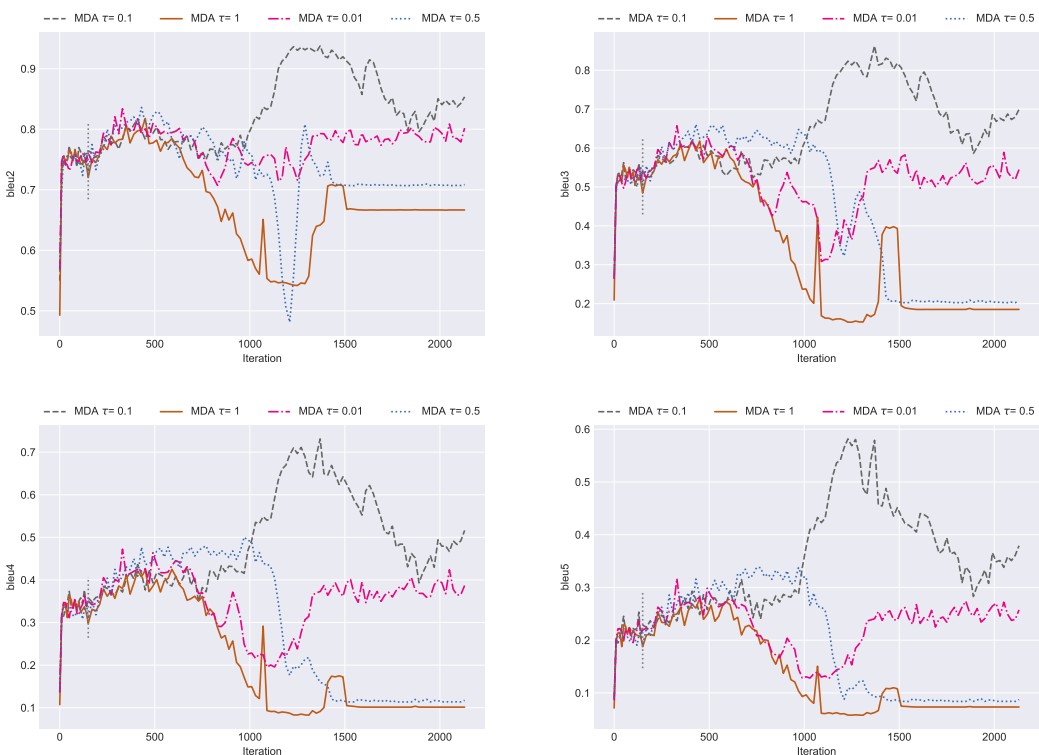

Figure 6: Training curves of BLEU scores on MS COCO Image Caption dataset with various Gumbel-Softmax temperature values.

Table 9: Randomly sampled 10 samples trained on MS COCO dataset, with MDA (top row) and MSA (bottom row). We can observe that the phrases "a man" and "motorcycle" occur too many times in the samples generated by MSA-enhanced models. Thus, models with MSA lack the diversity of text whereas those with MDA perform well on both the diversity and quality.

a woman sitting on a bench next to a bicycle parked near a sidewalk .
a person is riding a motorcycle on a city street with a woman standing on the back of it .
a cat sitting on a motorcycle on a car seat wanting to get out the car window .
a woman is sitting on a bench next to a bicycle .
a man is riding a motorcycle on a city street with parked vehicles .
a person is riding a motorcycle on a closed intersection with a group of people standing on the side of the road with people standing around it .
a woman is leaving a bathroom with a toilet and a sink .
a little girl is sitting on a bench next to a bicycle .
a little girl is sitting on the back of a storefront bathroom with his orange bike next to him .
a person sitting on a bench next to a bicycle parked near a wall .

a man is sitting on a motorcycle on a busy street , in a city .
a man is sitting on a motorcycle on a crowded street , near a building with a bike nearby .
a man is sitting on a motorcycle on a crowded street , near a bicycle .
a photo of a small plane sitting on a busy runway .
a cat sitting on a motorcycle on a crowded street near a parking lot .
a group of people sitting on a motorcycle on a side of a road .
a man is sitting on a motorcycle on a crowded street , near a street .
a person sitting on a motorcycle on a sidewalk next to a bicycle .
a man sitting on a motorcycle on a busy sidewalk next to a bicycle .
a man is sitting on a motorcycle on a busy street , in a city .

Table 10: Randomly sampled 10 samples trained on EMNLP2017 WMT News dataset, with MDA (top row) and MSA (bottom row).

---

the british people would have to commit to travelling to europe and has been a priority for the first time in a decade .
his priority finally becomes a hope for the police to be informed by the attack and is not on the scene .
there is more than a year before the start of the day after the european ' s first , but it was not in the last two years .
it is not something that is about to be a 15 . 6 percent in the fourth quarter , according to a report from the third of the week .
" i ' ve been a part of our last two years , " he said in a statement from the bbc ' s today .
now that ' s why we have to be a part of the UK .
i ' m not saying that was the first of the kind of people who were in the wrong but it is true that it is yet to be determined .
he will be a key for the first time in a decade , and has helped to stop the spread of the decade - over the past year .
so what if that ' s the reason that they can be within the last two years .
it was one of the most in the first quarter , but it was the first of the nearly two months since the start of the first of the day .

---

if you ' re a new , and you have to be a part of the team .
in a fox news , she has already been a major despite a conflict in the world .
when you ' re in the world , when they are growing .
but i ' ve been a part of the group for christmas .
instead , there is no evidence to suggest that the united kingdom .
the department of health and the enforcement and defense agencies last Thursday .
if you ' re the only in the world , and i ' m sure .
he will be more than it was a time when he was in the control of the water .
sometimes it was a number of the people who were in the back of the way .
it is an easy for trump ; the other at the same .

---

