# OpenReview forum: "Improving Sequence Generative Adversarial Networks with Feature Statistics Alignment"
_ICLR.cc/2021/Conference — Reject_

### Official Review · AnonReviewer4 · 2020-10-28
**Interesting idea but flawed experiments**

**Rating:** 4
**Confidence:** 3

**Review:**

**Main Claim:**

In this work, the authors propose to use the Feature Statistics Alignment paradigm to enrich the learning signal from the discriminator in a sentence generation GAN. The proposed model can generate sentences with better likelihood and BLEU on one synthetic and two real datasets.

**Contributions:**

This work introduces an novel and interesting idea of Feature Statistics Alignment in training GANs.

The authors follow the convention in this domain, and evaluate the model on three datasets.

The experiment results show that the proposed model outperforms existing models. However, the authors need to clarify some details to make the results trustworthy (see weakness).


**Strong points:**

The idea is novel and interesting.

The model and training procedure is clearly explained. Related works are cited well.


**Weak points:**

In Table 2:

- The LSTM model gets NLL lower than the real data. This is a clear evidence of overfitting.
- In SAL (Zhou et. al, 2020), NLL_{gen} is used to evaluate the diversity of the generator. But this metric is missing here without explanation.

In Table 3:

- The BLEU metric in this paper is the BLEU(F) metric in SAL (Zhou et. al, 2020). This metric evaluates the generated sentences using the test set as a reference. Thus the BLEU(F) metric cannot show the diversity of examples.
- The BLEU(B) (Zhou et. al, 2020) metric is missing. BLEU (B) metric evaluates the test set using the generated sentences as a reference, so it can detect mode collapse of a generative model.

In section 4, although authors clearly cite previous works for experiment settings, I think it’s worthwhile to repeat the definition of each metric, and some other key points in the paper, so that readers can easily understand the notations and jargons in this section.

**Recommendation:**
Reject.

There’s a major flaw in the evaluation metrics. On both synthetic and real datasets, the evaluation metrics prefers overfitted models, i.e. if the model can remember one example from the training set, and repeat that sentence, it can get a very high score.

I will reconsider my recommendation if (1) I miss interpret the metrics or (2) the authors provide more evidence on the diversity of the generated sentences, for example showing the NLL_{gen} metric on the synthetic dataset, and BLEU(B) metric on real datasets.

**Questions:**

How is NLL_gen computed?

**After Rebuttal**
The author's reply partially resolved my concerns, although the diversity of models has not improved, nor has it significantly decreased. Thus I have increased my score from 3 to 4.

---

> ### Author Response · Authors · 2020-11-21
> **Response to Reviewer4**
>
> Thank you very much for your comments. We address your specific questions and comments below:
>
> **Questions**
>
> 1."The LSTM model gets NLL lower than the real data. This is a clear evidence of overfitting."
>
> It is worth mentioning that the "real data" in Table 2 are also generated by an LSTM model as the oracle. We were also surprised that the vanilla LSTM could achieve the lowest NLL score when the generated sentence length is 20. However, it performed not as well as the RMC when generating long sentences (with a length of 40). We guess this is due to the fact that the oracle model of synthetic data is a single layer LSTM, which could result in the **LSTM-biased** phenomenon for short text generation (length 20). As for long sentence generation, since the vanilla LSTM could forget some information with a long-term span, which could lead to the performance drop as reported in Table 2. We show that the RMC performs better than LSTM for generating long sequences.
>
> 2."NLL_{gen} is missing."
>
> Yes, NLL_{gen} could be used to measure the diversity of generated sequences. In our experiments, we just use the synthetic data to validate the effectiveness of our method and do not report the NLL_{gen} metric in the submitted version (following SeqGAN, RankGAN, LeakGAN). Instead, we use NLL_{gen} to measure the generation diversity on real datasets, as shown in Table 3.
>  For clarity, we have reported the NLL_{gen} scores in Appendix C.1 of the revised version. The NLL_{gen} score of our model is similar to that of baseline models.
>
> 3."BLEU(F) metric cannot show the diversity of examples" and "BLEU(B) (Zhou et. al, 2020) metric is missing"
>
> Yes, BLEU(F) is used to reflect the **quality** of samples, whereas NLL_{gen} is applied to indicate the **diversity** of generated sentences. The adoption of BLEU(F) and NLL_{gen} follows the previous work, such as Texygen(Zhu et al., 2018) and RelGAN (Nie et al., 2019). To answer Reviewer4's question, we report the BLEU(B) as the following table. We found that the proposed method achieves similar results in terms of the diversity compared with SAL (Zhou et al., 2020): on the MS COCO dataset, the BLEU(F) score of our method is a bit lower but got better NLL_{gen}; on EMNLP2017 WMT News dataset, our method achieves slightly better than SAL, but a bit worse on NLL_{gen}. But our method achieves a significant improvement on BLEU(F) metrics.
>
> On the MS COCO dataset:
>
> |             |   Bleu-2 (B) |   Bleu-3 (B) |   Bleu-4 (B) |   Bleu-5 (B) |  NLL_{gen} |   Bleu-2 (F) |   Bleu-3 (F) |   Bleu-4 (F) |   Bleu-5 (F) |
> |-------------|--------------|--------------|--------------|--------------|------------|--------------|--------------|--------------|--------------|
> |  SAL        |  **0.724**       |  **0.503**       |  **0.313**      |  **0.198**       |  0.873     |  0.785       |  0.581       |  0.362       |  0.227       |
> |  Ours (MSA) |  0.700       |  0.404       |  0.206       |  0.123       |  0.760     |  **0.959**       |  **0.866**      |  **0.759**       |  **0.630**       |
> |  Ours (MDA) |  0.712       |  0.413       |  0.209       |  0.121       |  **0.717**    |  0.938       |  0.863       |  0.731       |  0.582       |
>
> On the EMNLP2017 WMT News dataset:
>
> |             |   Bleu-2 (B) |   Bleu-3 (B) |   Bleu-4 (B) |   Bleu-5 (B) |  NLL_{gen} |   Bleu-2 (F) |   Bleu-3 (F) |   Bleu-4 (F) |   Bleu-5 (F) |
> |-------------|--------------|--------------|--------------|--------------|------------|--------------|--------------|--------------|--------------|
> |  SAL        |  0.726       |  0.431       |  0.232       |  0.123       |  **2.578**     |  0.788       |  0.523       |  0.281       |  0.149       |
> |  Ours (MSA) |  0.742       |  **0.480**       |  **0.252**       |  **0.138**       |  3.999     |  **0.932**       |  **0.798**       |  0.585       |  **0.404**      |
> |  Ours (MDA) |  **0.744**       |  0.474       |  **0.252**       |  0.137       |  2.732     |  0.916       |  0.784       |  **0.592**       |  0.386       |
>
> 4."How is NLL_gen computed?"
>
> NLL_{gen} computes the neg-log likelihood of reference samples in the test set by the generator, which is adopted by SAL (Zhou et. al. 2020) and RelGAN (Nie et. al. 2019) to measure the diversity of generated samples. We have clarified this in the paper.
> To sum up, we use NLL_{oracle} and BLEU(F) to automatically evaluate the sample quality and employ the NLL_{gen} to evaluate the diversity of generated samples. The proposed model achieves superior performance than previous work, and remain similar diversity of generated sentences.
>
> Thank you for pointing out the issues of evaluation metrics, and **please let us know if we could address your concerns about evaluation.**

---

### Official Review · AnonReviewer2 · 2020-10-28
**Official Blind Review #2**

**Rating:** 6
**Confidence:** 3

**Review:**

Summary:

The paper addresses the task of improving GANs for sequence generation and proposed a method based on the relativistic discriminator. The proposed method employs a Feature Statistics Alignment (FSA) paradigm to reduce the gap between real and generated data distributions. It relies on the relativistic discriminator for "coarse" differences and FSA for "fine-grained" differences between real and generated data distributions. It is evaluated on synthetic and real datasets, and it significantly outperforms the baselines. It also outperforms baselines on human evaluation based on the acceptance, grammaticality, and meaningfulness of the generated sentences.

Strengths:

The proposed approach is very effective, as demonstrated by significant performance improvements in the experiments across synthetic and real datasets. Also, it can generate better sentences compared to the baselines, as shown by human evaluation.

Weakness:

Although the proposed model is thoroughly evaluated and empirically effective, it is not very different from existing methods, except for FSA. The application of FSA in this context might be novel; however, the proposed approach seems to be a simple combination of two existing approaches. Therefore, the novelty of the model is limited.

Minor comments:
1. Correction:  Did you mean

In contrast, the lower τ could discourage exploration and tend to explore during training. -> In contrast, the lower τ could discourage exploration and tend to exploit during training. ?

2. It would have been good to see a head-on comparison of the generated samples (baseline vs. proposed approach) in the paper's main text.

---

> ### Author Response · Authors · 2020-11-21
> **Response to Reviewer2**
>
> Thank you for your detailed comments and reviews. We address your specific questions and comments below:
>
> Questions:
>
> 1.About novelty:
>
> **A short answer**: Different motivation and usage. Our paper aims to mitigate the difficulty of the generator training by providing **additional signals** from the discriminator, whereas conventional feature matching serves as the training objective for both the generator and discriminator in language GANs. The FSA technique empirically achieves superior performance on Gumbel-Softmax GANs without introducing extra training parameters.
>
> **Detailed answer**:
> The paper proposed an FSA mechanism to improve the Gumbel-Softmax-based GAN's training, and achieves the superior performance compared with previous works. It is indeed that the FSA is a form of feature matching. However, previous language GANs using feature matching directly optimize the feature matching metrics, such as TextGANs, FM-GANs, etc. Instead of directly optimizing the feature matching loss for both the generator and discriminator, we use FSA to align the feature distribution and assist the training of only the generator. Besides, the proposed FSA is used to provide fine-grained learning signals for the generator besides the original (coarse) learning signals of language GANs. The idea of FSA could also be regarded as a resemblance to feature leakage in LeakGAN (Guo et. al. 2018), which leaks the features from the discriminator to the generator.
>
> 2.Thank you for pointing out the mistake, it should be "the lower τ could discourage exploration and tend to **exploit** during training".
>
> 3.Due to the page limit, we do not include the comparison between generated samples and the proposed method. We added it in Table 8 (Appendix E.1) of the revised version:
>
> |            | Samples                                                                                                                                                               |
> |------------|-----------------------------------------------------------------------------------------------------------------------------------------------------------------------|
> | Real       |(1) a single kite flies high above a body of water as a person stands on the edge of the water .  (2) a man wearing an apron in an industrial kitchen reaching for a pot .     |
> | MLE        |(1) a man watches on his bike , in a lake on a field . (2) a women is standing behind an orange table in helmet on a child in the background .                                |
> | SeqGAN     |(1) some people sitting on top of luggage near a truck . (2) a man sitting in a bath tub on tops .                                                                            |
> | TextGAN    |(1) a man riding a motorcycle . (2) a bathroom with a sink , and a table .                                                                                                    |
> | LeakGAN    |(1) a man standing next to her cell phone on a street sign . (2) a woman is holding a child in the air .                                                                      |
> | MaliGAN    |(1) a woman is standing and another oak cake on a drain . (2) a man standing in a kitchen with her laptop and two tables                                                      |
> | RankGAN    |(1) a colorful bike is is down next to a large mirror .  (2) a man is riding a bike down a track .                                                                             |
> | RelGAN     |(1) a woman walking with a dog in the city in front of a city bus .  (2) a man sitting on a bed in a room with a chair on the couch .                                          |
> | Ours (MSA) |(1) a man is sitting on a motorcycle on a busy street , in a city .  (2) a man sitting on a motorcycle on a crowded street near a building , with a bicycle in a parking lot . |
> | Ours (MDA) |(1)  a person is riding a motorcycle on a city street with a woman standing on the back of it .  (2) a man with a woman standing next to a fire hydrant wearing a backpack .    |
>
>
>
> We hope we could address your concerns. **Please feel free to let us know if you still have questions.**

---

### Official Review · AnonReviewer1 · 2020-10-28
**Interesting paper**

**Rating:** 6
**Confidence:** 3

**Review:**

[Summary]
This paper proposes a new GAN-based text generation method that incorporates feature statistics alignment and gumbel-softmax for reparameterization to deal with mode collapse and unstable training. For feature statistics alignment, the authors design two methods such as mean square and mean distance alignments. They evaluate the proposed method on a synthetic dataset, MS COCO caption, and EMNLP2017 WMT news dataset, comparing them with RL-based and non RL-based models. With extensive experiments including ablation studies, the proposed method show promising results.

[Recommendation]
Overall, this paper is clear and well-written. So I lean to acceptance. But I have some concerns as well.

[Strength]
- Mode collapse is challenging issue in GAN training.
- Text generation is important problem.

[Weakness]
- The authors insist the use of Gumbel-softmax in GAN tranining is under-explored. But It is not clear. There are more method using Gumbel-softmax [Gu et al. 2019] and a similar softmax with temperature annealing. It is not clear for the authors to explicitly discriminate using Gumbel-softmax and other smoothed softmax methods.
- Some related work  were missed such as DialogWAE [Gu et al. 2019] and ARAML [Ke et al. 2019]. In particular, DialogWAE uses GAN and Gumbel-softmax for text generation even if it focuses on dialog generation.
- For verifying mode collapse issues, how about using Self-BLUE in addition to BLUE scores as a metric to evaluate the diversity?
- Novelty might be incremental. It seems that the novelty is from using feature statistics alignment. To emphasize the contribution of feature statistics, comparing between the latent feature visualization with and without FSA might be helpful in addition to ablation study.

[Minor]
In p3, the given real data is --> are


[Gu et al. 2019] DialogWAE: Multimodal Response Generation with Conditional Wasserstein Auto-Encoder. ICLR 2019.
[Ke et al. 2019] ARAML: A Stable Adversarial Training Framework for Text Generation. EMNLP 2019.

---

> ### Author Response · Authors · 2020-11-21
> **Response to Reviewer1**
>
> Thank you for your helpful reviews. We address your specific questions and comments below:
> ## Question #1
> We agree that there are several works in language GANs that employ the temperature annealing in the generator. However, there are only few works, such as RelGAN (Nie et.al. 2019), that leverage Gumbel-Softmax tricks in language GANs on real datasets. By saying "it is under-explored", we mean that there remains an open problem on its improvement. Thank you for pointing out this, we have clarified it in the revised paper.
> ## Question #2
> Thank you for giving the related work of language GANs on dialogue generation. We have added these works on the paper.
> ## Question #3
> Good question! Indeed, the Texygen benchmark (Zhu et al., 2018) we followed used Self-Bleu to measure the diversity of generated real sentences. However, it is not feasible to use it as the evaluation metric.
>
> **A short answer**:
>
> There is a problem when using the official implementation of Self-Bleu, which has been pointed out in previous work (Nie et. al. 2019; Zhou et. al. 2020).
>
> **Detailed answer**:
>
> Previous work did not use it since there is an issue in the official implementation (https://github.com/geek-ai/Texygen/blob/master/utils/metrics/SelfBleu.py): the generated sentences changed but the reference remained the same during the evaluation process. It is discussed in the appendix of SAL(Zhou et. al. 2020) and the openreivew of RelGAN paper (Nie et. al. 2019). As reported in SAL and RelGAN, our self-BLEU (2-5) scores are also always 1. Thus we do not use the Self-BLEU metric.
>
> For reference, we copy it here:
>
> '''Note that many previous works use self-BLEU Zhu et al. (2018) as a diversity metric. However, we find that there exists a problem in the official implementation of the self-BLEU metric: Only in the first time of evaluation that the reference and hypothesis come from the same “test data” (i.e. the set of generated sentences). After that, the hypothesis keeps updated but the reference remains unchanged (due to “is-first=False”), which means hypothesis and reference are not from the same “test data” anymore, and thus the scores obtained under this implementation are not self-BLEU scores. To this end, we modified the implementation to make sure that the hypothesis and reference are always from the same “test data” (by simply removing the variables "self.reference" and "self.is-first") and found that the self-BLEU (2-5) scores are always 1 when evaluating all the models. This problem is also discussed in the openreview of the RelGAN paper.''' (Zhou et. al. 2020)
>
> ## Question #4
> Good point! Yes, it would be helpful to add a TSNE visualization for latent features with and without FSA. We appreciate your notification and would try to add it in the future version.
>
> Thank you for your helpful suggestions and comments! **Please let us know if you still have some concerns or questions.**

---

### Official Review · AnonReviewer3 · 2020-10-29
**The paper proposes an improvement to sequence generative adversarial networks (GAN) by combining Gumbel-Softmax based GAN with the matching of mean representations of true and generated samples in a latent feature space. Experimental evaluations on synthetic and real datasets show the effectiveness of the method.**

**Rating:** 5
**Confidence:** 3

**Review:**


Summary:

The paper proposes an improvement to sequence generative adversarial networks (GAN) to cope with the common training issues of GANs. For the sake, the paper combines Gumbel-Softmax based GAN, relativistic discrimination  function with  the matching of mean representations of true and generated samples in a latent feature space. This feature statistics alignment allows to leak information from the discriminator to the generator as the used features are extracted from the discriminator network. Experimental evaluations on synthetic and real datasets show the improvement achieved by the proposed method over existing sequence generation networks.



Reasons for score:

The paper straightforwardly combines existing procedures (relativistic discriminator, Gumbel-Softmax approximation for categorical distribution, features matching) to improved upon vanilla sequence generation networks and somehow lacks novelty. Although the ablation study is interesting and shows the improvement brought by each module, examples of lengthy generated sequences illustrate that the sentences produced by the GAN are not semantically meaningful.


Pros:
- Overall, the paper is well written. In particular, the rationale behind the proposed method is justified. Empirical evaluations support these intuitions and show how they contribute to the observed quality of the generated sequences.
- The paper aims at addressing a major issue in training GAN for sequence generation: how to strengthen the learning of the generator compared to the discrimination network which is easier to train? The approach promoted in the paper consists to align the mean statistics of true sequences and fake ones. Specifically, the statistics are computed over features extracted from discrimination network. The objective function of the generator is therefore composed of the usual GAN loss term and the distance between those mean representations. As such, this idea of guiding the generation network with information from the discriminator is interesting and plays a key role in the performances improvement.
- In the same vein, the use of Gumbel-Softmax distribution (instead of the discrete distribution) and of the relativistic discrimination function (instead of the classical classification function) helps to learn a better generation model. However these ideas are not novel and were investigated separately in previous research works.
- Experimental evaluations, including both qualitative analysis and quantitative results, are provided in the paper and in the supplementary to show the effectiveness of the proposed framework. The newly proposed GAN achieve superior performances. The comprehensive ablation study is interesting and helps to understand how each module (feature alignment, Gumbel-Softmax, batch size) contributes to the enhanced performances.

Cons:
- Although the proposed method, according to the empirical results, show improved performances, it lacks novelty as features matching, relativistic discrimination or Gumbel-Softmax are not new ideas. The main contribution resides in the better quantitative results compared to existing sequence generation networks. However when one examines the generated sentences, it appears that they lack semantic meaningfulness especially for long sentences (see for instance Table 8). This shows that the proposed GAN (as well as the competitors) is not effective yet.
- Features distribution alignment is an interesting way to measure how close are the marginal distributions of the real and fake sequences. The paper considers the Mean Distance Alignment (MDA) and the Mean Square alignment (MSA) which are respectively the distance and the squared distance between the mean latent representations of the real and generated sequences. Several comments can be made as hereafter.
     * MSA and MDA encode the same matching up  to a power 2. It’s unclear why they lead to different empirical results.
     * Instead of matching only the mean statistics, the overall distributions of the latent representations can be aligned by considering metrics such as MMD or Wasserstein distance. How would the results look like in that setting?
     * It should be clarified earlier in the paper that the used features are extracted from the discrimination network (as the weights between the discriminator and feature extractor are shared). Also the paper should make explicit from which layer of the discrimination network the features are extracted.
- The findings of human evaluation (see Table 5) are not unequivocal. MSA and MDA achieve higher scores than the real sentences. The best model, the one with MSA, is not preferred because of a lack of diversity and quality. This raises the question of how reliable is the human evaluation score.

Other comments:
- Page 3, definition of MSA: in the sentence “mean squared difference between the centroids...”, the term mean is over-used as Eq. (2) or (3) represents only the squared distance between centroids.
- Page 4: in “The FSA term on the RHS can also be regarded as a dynamic regularizer for the sequence generator” the notion of dynamic regularizer is unclear. In which sense FSA induces a dynamic regularization?
- In Equation (10) the function “one_hot” should be defined. Also in (10), I think $y_i$ should read the $|V|$-dimensional vector $y$.
- Equation (11) is to be checked carefully as the parameter $\tau$ simplifies in numerator and denominator.
- Algorithm 1: the update of the generator $G_\theta$ requires a minibatch from real dataset in order to minimize $L_{RG}$ + $L_{FSA}$ as $L_{FSA}$ relies on the mean of real data latent representation.

After rebuttal
-  I read the response of the authors. The spotted typos are fixed in the revision. Some  questions/concerns  have been tentatively. However the novelty in the paper is still not blatant or how the use of distance such as MMD or Wasserstein to match the features is under-explored. Hence I intend to keep my rating.

---

> ### Author Response · Authors · 2020-11-21
> **Response to Reviewer3 (Part 2)**
>
>
> 8."MSA and MDA achieve higher scores than real sentences."
>
> This is because MSA and MDA tend to generate longer grammatically correct sentences, whereas the MSA tends to fall into limited patterns (but still with good quality). The quality of reference captions in MS COCO Image Caption datasets is various, in which some sentences would be single phrases. It makes sense to get a higher score if the model tends to generate sentences with a longer length and better grammaticality.
>
> 9."The term mean is over-used"
>
> Thank you for pointing out this detail! We rewrote the description before Eq.(2) for clarity.
>
> 10."In which sense FSA induces a dynamic regularization?"
>
> From the initial idea of our paper, the FSA serves to provide fine-grained learning signals during the training process, besides the original GAN's signal. From the perspective of loss terms, the FSA could be regarded as a constraint that dynamically modulates the feature alignments between real and generated feature representations. To avoid misinterpretation, we have removed this claim.
>
> 11."In Equation (10) the function “one_hot” should be defined. y_i should read the |V|-dimensional vector y"
>
> Thank you for your carefulness. We have modified this in the revised paper.
>
> 12."Equation (11) and Algorithm 1".
>
> We have modified them according to your paper. Thank you very much for the comments!
>
> We appreciate for providing the detailed comments and have modified them according to your reviews. **Please feel free to let us know if we could address your concerns**.

---

> ### Author Response · Authors · 2020-11-21
> **Response to Reviewer3 (Part 1)**
>
> Thank you for your helpful reviews. We address your specific questions and comments below:
>
> Questions:
>
> 1."It lacks novelty as features matching, relativistic discrimination or Gumbel-Softmax are not new ideas."
>
> **A short answer**: Different motivation and usage. Our paper aims to mitigate the difficulty of the generator training by providing **additional signals** from the discriminator, whereas conventional feature matching serves as the training objective for both the generator and discriminator in language GANs. The FSA technique empirically achieves superior performance on Gumbel-Softmax GANs without introducing extra training parameters.
>
> **Detailed answer**:
> The paper proposes the FSA mechanism to improve the Gumbel-Softmax-based GAN's training and achieves superior performance compared with previous works. It is indeed that the FSA is a form of feature matching. However, previous language GANs using feature matching directly optimize the feature matching metrics, such as TextGANs, FM-GANs, etc. Instead of directly optimizing the feature matching loss for both the generator and discriminator, we use FSA to align the feature distribution and assist the training of only the generator. Besides, the proposed FSA is used to provide fine-grained learning signals for the generator besides the original (coarse) learning signals of language GANs. The idea of FSA could also be regarded as a resemblance to feature leakage in LeakGAN (Guo et. al. 2018), which leaks the features from the discriminator to the generator.
>
> 2." it appears that they lack semantic meaningfulness especially for long sentences (see for instance Table 8)"
>
> Yes, there may be some unmeaningful sentences in the randomly generated samples, which is common in most language GANs because it is difficult for generating long sentenced. We resampled the generated samples with the model of the best quality. Please see Table 10 in Appendix E.2.
>
> 3."MSA and MDA encode the same matching up to a power 2. It’s unclear why they lead to different empirical results."
>
> We empirically find that MSA prefers to generate longer sentences but may fall into the mode dropping. We guess this is because of the gradient difference between MDA and MSA. For MDA, the gradient of an absolute value is $\pm 1$, which is a bit difficult to converge to the local optimum. In contrast, the gradient of MSA has a more flexible range of values: it gets smaller when the predicted loss becomes smaller, pushing it faster to converge to the local minimum. As a result, the convergence on the local minimum may not perform very well on diversity.
>
> 4."Using MMD or Wasserstein distance"
>
> Good point! It is a great idea to investigate whether feature matching metrics can be beneficial when substituting FSA. Our initial idea is to align the lower-degree statistics of feature representations. The exploration of unified feature matching methods can be left for future work.
>
> 5."It should be clarified earlier in the paper that the used features are extracted from the discrimination network (as the weights between the discriminator and feature extractor are shared)"
>
> We moved this part to Sec. 3.1.
>
> 6."Which layer of the discrimination network the features are extracted"
> As in Sec. 3.2, we denote H(.) as the non-transformed layer before the non-linearity. Thus the feature extractor H(.) is the last layer before the activation function. We have made it more clear in Sec. 3.2,
>
> 7."Question of how reliable is the human evaluation score"
>
> Human evaluation is used to measure the quality (i.e., acceptance, grammaticality) rather than the diversity. We perturbed the sentences and anonymized the model's identity before the evaluation.

---

### Author Response · Authors · 2020-11-24
**Hi reviewers, we are looking forward to your reply!**

Hi reviewers, thank you for your comments and helpful advice! Please note that the discussion period would be due on 24 Nov and we would not respond afterward. We would like to discuss if you still have concerns or comments.

---

### Decision · Program_Chairs · 2021-01-07
**Final Decision**

**Decision:**

Reject

**Comment:**

The work introduces a method that uses the Feature Statistics Alignment paradigm to improve sequence generation with GANs. The contribution is interesting and novel (although marginally), clarity is also good.
However the reviewers raised several concerns calling for more comprehensive and thorough evaluation. Experiments show an improvement comparing to selected baselines and the revised paper addressed, at least partially, a serious evaluation concern of one reviewer.
Although the excellent revision work some important open questions still seem to remain, in particular the choose of alignment metrics and a thorough evaluation.